# Effects of Blended Learning Program for Cyber Sexual Harassment Prevention among Female High School Students in Bangkok, Thailand

**DOI:** 10.3390/ijerph19138209

**Published:** 2022-07-05

**Authors:** Siriporn Santre, Tepanata Pumpaibool

**Affiliations:** College of Public Health Sciences, Chulalongkorn University, Bangkok 10330, Thailand; ss.santre@gmail.com

**Keywords:** cyber sexual harassment, blended learning, female high-school students, Thailand

## Abstract

Cyber sexual harassment has been increasing and has become a major public health problem among youths. Therefore, this study primarily aimed to evaluate the effects of a blended learning program on knowledge, attitudes, intention to cope with cyber sexual harassment, behavioral coping strategies, and cyber sexual harassment among female high-school students in Bangkok, Thailand. A quasi-experimental study with a two-group design was conducted from May 2021 to October 2021. A total of 112 students were recruited into either an intervention (*n* = 56) or a control (*n* = 56) group. The intervention group participated in a blended learning program for 12 weeks. A self-administered questionnaire was completed by both groups at pre-intervention, post-intervention, and follow-up. Chi-square test, Fisher’s exact test, and repeated measures analysis of variance (ANOVA) were used for data analysis. The mean score of the intervention group in knowledge, attitude, intention to cope with cyber sexual harassment, behavioral coping strategies, and cyber sexual harassment after completing the program and follow-up were significantly different from baseline and the control group (*p* < 0.05). A positive effect of the blended learning program was observed among female students. Therefore, this program can be useful for victims, supporting their self-confidence with decreased frequency of cyber sexual harassment experiences.

## 1. Introduction

Online or cyber harassment has increased in recent years to become a major public health issue among adolescents [1]. Digital interactions such as exploring sexuality or building relationships play an important role in the lives of young people and may potentially result in risky or harmful online behaviors [2,3]. Digital platforms, including social networking sites, are heavily used by youths and present new methods of sexual harassment [4,5]. According to prior studies, females are more likely to report being objectified owing to their gender and to encounter sexual harassment online more often than other groups [6,7]. Female students are the group most at risk since males are more likely to engage in online sexual harassment as harassers [7].

The prevalence of cyber harassment and other forms of gender-related violence online has become increasingly harmful [8]. In Egypt, 79.8% of female students had experienced cyber sexual harassment at least once throughout the past 6 months [9]. Approximately 4 of 10 American adults have experienced online harassment, and 18% of them have been the target of severe behaviors [10]. Among Croatian female adolescents, 37.5% of 477 students had reportedly experienced cyber sexual harassment. Adolescents who experience cyber sexual harassment suffer negative impacts, particularly in terms of their mental health and health-related behaviors. The victims show anxiety, depression, poor academic performance, and absenteeism from school [11].

In Thailand, the use of smartphones, the Internet, and social media is increasing [12]. Nearly 93% of students aged <20 years used social media and the estimated average duration of daily Internet usage was 12 h a day. Facebook was the most popular social media among adolescents [13]. Approximately 49.3% of the students aged between 15 and 24 years from a study in Central Thailand reported online harassment and cyberbullying [14]. A survey from the department of children and adolescents found that approximately 4000 Thai youths had met online friends at least once and admitted having been cyber sexually harassed or cyberbullied by strangers one way or another [15]. Another previous survey conducted among Thai students from 15 schools nationwide reported that almost 30% of 120 students had experienced some type of abuse or harassment on social media [16]. Cyber sexual harassment is often under-reported as perpetrators consider it “normal” and some victims choose to ignore it [17]. In addition, in comparison to face-to-face sexual harassment, only a few studies have been conducted on cyber sexual harassment [18].

Several prevention methods for cyber sexual harassment have been implemented, including providing education, improving awareness, changing norms, empowering students, and seeking social support [7,19,20]. Only a few students consulted their parents to deal with cyber harassment because they consider that adults are unaware of these issues and they are afraid that their parents will limit their Internet usage. However, most students chose to keep it a secret and consulted with their peers [21,22,23]. Despite the growth of cyber sexual harassment, the theoretical background explaining such online behaviors has not been widely investigated; however, utilizing the theory of planned behavior and empowerment would more accurately pave a way that may lead to the decreasing the incidence of the problem. An educational intervention compiled with theory may improve the behaviors among students [24].

Students’ attitudes were associated with their intention to perform such behaviors, which may not be changed due to inadequate knowledge and poor attitude. A significant factor in the theory of planned behavior is the intention to perform (or not to perform) a certain behavior. Although a variety of factors influence intentions, most research does not utilize a behavioral theory [25]. This study uses the theory of planned behavior to increase victims’ intentions and behavior in coping with their victimization experiences. Moreover, there is a wide range of needs among victims in terms of empowerment and support. Such needs must be provided through a well-managed and integrated strategy for the powerless victim to recover from harassment exposure. Empowerment theory is also associated with an individual’s well-being and competencies and helps provide or increase power among victims, instead of blaming them. Female victims of sexual and gender-based harassment are frequently targeted by empowerment programs, with the rationale that these women may develop disempowered beliefs as a coping strategy [26,27].

Furthermore, some of what is already known regarding online sexual harassment derives from in-person sexual harassment research and other related online behaviors [18,28]. Blended learning improves student achievement and satisfaction, and it has advantages for students, including usefulness, flexibility, and effectiveness. Additionally, instructors may provide students instant access to their course materials from any place and at any time. Students can review what they have learned through an online platform. This improves learning retention and engagement among students [29]. Thus, this study aimed to develop a blended learning program to prevent cyber sexual harassment. The effects of this program on knowledge, attitude, intention to cope with cyber sexual harassment, behavioral coping strategies, and cyber sexual harassment among female high-school students in Bangkok, Thailand, were evaluated.

## 2. Materials and Methods

### 2.1. Study Design and Sampling

Quasi-experimental research with a two-group design was conducted. The intervention group received a blended learning program, whereas the control group did not receive any program. The intervention program was conducted from May 2021 to October 2021. To determine the sample size, power analysis with a power of 0.80 (α = 0.05, β = 0.20) was performed [30]. The calculated sample size in each group was 56 students, with 20% added to prevent the effects of dropout and missing information in data analysis. Therefore, a total of 112 students were included.

To recruit students, a multi-stage sampling technique was applied. Two high schools in Bangkok under the supervision of the Office of the Basic Education Commission were selected by simple random sampling based on their similar characteristics, such as their curriculum, size of school, and reporting of cyberbullying as high-level online harassment. Female students in grade 10 were selected using purposive sampling for both the intervention and control schools. Finally, simple random sampling was used to select the female students. The inclusion criteria were students who experienced cyber sexual harassment in the last 6 months, had a smartphone and Facebook account, and could use Facebook’s features by themselves. Students with mental and physical health problems and those who missed classes in more than two sessions during the program were excluded. Permission from students and their parents was obtained using a consent form.

### 2.2. Intervention

The blended learning program consisted of 12 activities and was carried out for 12 weeks. Each session included classroom and online learning. Classroom activities took 90 min every week with Facebook assignments and games. Before starting the activity, students were divided into six subgroups. The reason for group gathering or group process was that students could have positive interactions with each other and support their peers. Four research assistants were trained and helped to run the program in-class and online. A private Facebook group (closed group) was created as a part of the online learning. Using this platform, students could learn content related to each in-class activity whenever and wherever they wanted to. Group discussions, lectures, games, and the sharing of personal experiences all took place throughout the class sessions. Assignments, questions and answers, online tests, videos, and opinion exchanges on tasks at any time were all used in online activities. The information was only accessible to the participants and the researcher team. Students’ privacy and information were carefully protected. The program contents are shown in Table 1.

### 2.3. Measures

The measurement tools were a self-administered questionnaire consisting of seven parts, including (1) sociodemographic characteristics, (2) Internet usage, (3) knowledge regarding cyber sexual harassment, (4) attitude toward cyber sexual harassment, (5) intention to cope with cyber sexual harassment, (6) behavioral coping strategies, and (7) number of cyber sexual harassment occurrences. The content of the questionnaire was validated by five experts. The item objective congruence index of all items was within the range of 0.74 to 0.91. The reliability of the questionnaire was assessed for knowledge regarding cyber sexual harassment (internal consistency reliability was 0.77), attitude toward cyber sexual harassment (a 5-point Likert scale was used to rate respondents’ agreement with each item and Cronbach’s alpha was 0.79), intention to cope with cyber sexual harassment (a 5-point Likert scale was used to rate respondents’ agreement and Cronbach’s alpha was 0.86), behavioral coping strategies (a 3-point Likert scale was used to rate respondents’ agreement and Cronbach’s alpha was 0.89), and cyber sexual harassment (a 4-point Likert scale was used to rate the experience of cyber sexual harassment among students in the past 6 months and Cronbach’s alpha was 0.87). The questionnaire was provided to both control and intervention groups at baseline, week 12, and week 20 of the study.

### 2.4. Data Analysis

Data analysis was carried out using Statistical Packages for the Social Sciences version 22.0. The significant level was set at a *p* < 0.05. Chi-square and Fisher’s exact test were used to analyze the difference between intervention and control groups on sociodemographic characteristics and Internet usage in categorical data. A repeated measures ANOVA was used to evaluate the effects of the program with the difference in mean scores of knowledge regarding cyber sexual harassment, attitude toward cyber sexual harassment, intention to cope with cyber sexual harassment, behavioral coping strategies, and number of cyber sexual harassment occurrences among the three time periods.

### 2.5. Ethical Considerations

Ethical approval was obtained from the Ethics Review Committee for Research Involving Human Research Subjects, Health Science Group, Chulalongkorn University (COA No. 065/2564). Before starting the intervention, parents willingly consented to their children as study participants by signing an informed consent form.

## 3. Results

A total of 56 students in the intervention group and 56 students in the control group were included. The results are presented in three parts: (1) sociodemographic characteristics, (2) Internet usage, and (3) effects of blended learning program. At baseline, the majority of students in the intervention (51.8%) and control groups (48.2%) had a grade point average between 2.01 and 3.00. Most of the students were heterosexual: 85.7% and 91.1% in the intervention and control groups, respectively. The majority of the students were single and stayed with their parents. Their parents’ marital status was married or living together. No statistically significant difference was observed on the sociodemographic data between two groups at baseline (*p* > 0.05) as shown in Table 2.

More than half of the students in the intervention (53.6%) and control groups (66.1%) spent more than 7 h a day surfing the Internet. Approximately 72% and 63% of the students in intervention and control groups, respectively, used Facebook. They usually used the Internet between 18:01 and 21:00 h without their parents’ supervision. At baseline, the Internet usage characteristics of both groups were similar. There was no statistically significant difference observed between the two groups (*p* > 0.05), as shown in Table 3.

Students’ knowledge, attitude, and intention to cope with cyber sexual harassment, behavioral coping strategies, and experiences of cyber sexual harassment were assessed at the end of the program (week 12) and 2 months after the program ended (week 20). The comparison of the mean scores using repeated measures ANOVA revealed an interaction effect between group and time. There were statistically significant differences between the intervention and control groups for all five outcomes i.e., knowledge (*p* = 0.028), attitude (*p* = 0.014), intention to cope with cyber sexual harassment (*p* = 0.016), behavioral coping strategies (*p* = 0.031), and occurrence of cyber sexual harassment (*p* = 0.025), after completing the 12-week program and 2 months after the program ended. The mean knowledge score showed a small but statistically significant (*p* = 0.014) increase in the intervention group from baseline to post-test and at follow-up (6.61 ± 1.86 to 7.36 ± 1.53 and 7.41 ± 1.56, respectively). Comparing the mean knowledge scores between the control and intervention groups at each assessment time, the intervention group showed a higher mean score at both time points (*p* = 0.022). Regarding students’ attitude toward cyber sexual harassment prevention, the intervention group’s score slightly increased after the intervention and this increase persisted at the follow-up assessment (38.36 ± 5.33 at baseline to 40.43 ± 4.73 and 40.46 ± 4.65 at post-test and follow-up, respectively; *p* = 0.017). The increase in the score for the intervention group was higher than that for the control group at the same assessment time (*p* = 0.003).

In terms of intention and behavioral coping strategies to prevent cyber sexual harassment, the mean score of the intervention group was slightly increased from baseline to post-test and follow-up with values of 39.43 ± 5.26, 41.34 ± 4.62, and 41.84 ± 4.93, respectively (*p* = 0.003) for the intention outcome and 23.30 ± 2.18, 24.25 ± 2.02, and 24.14 ± 2.08, respectively (*p* = 0.004) for behavioral coping strategies. In addition, the mean scores for intention and behavioral coping strategies of students in the intervention group were higher than those of students in the control group at the end of the program and at follow-up (*p* = 0.042 for intention and *p* < 0.001 for behavioral coping strategies). The blended program also influenced the occurrence of cyber sexual harassment, with the mean score of the intervention group decreasing after completion of the program and persisting at follow-up (*p* < 0.001), as shown in Table 4.

## 4. Discussion

This study used a quasi-experimental research design to evaluate the effects of a blended learning program for cyber sexual harassment prevention among female high-school students. The program is based on two theories that support the contents of the program. First, the empowerment theory helps adolescents exert more strength than their harassers, which is related to strategies and resources that encourage females to escape from negative situations in their lives and to assist students in improving their lives, avoiding inappropriate use of new technology, and increasing understanding and techniques to protect themselves on the Internet. Some studies have implemented their programs based on the concepts of the empowerment theory [30] and the planned behavior theory [25] with success, emphasizing attitudes, intentions, and behaviors to cope with the students’ cyber sexual harassment.

There was a significant difference between the intervention and control groups, as well as between the times of measurements on knowledge, attitudes, intention, behavioral coping strategies, and cyber sexual harassment. This could be due to using different methods of educational program, such as lectures, videos, presentations, games, quizzes, and group discussion-based programs. The knowledge scores of the intervention groups appeared to improve from the baseline to the follow-up assessment. The follow-up test revealed that this knowledge was more consistently retained through week 20. Thus, it is likely that the lesson on definition, risk factors, and prevention strategies helped students to more consistently retain knowledge and improved awareness of the significance of understanding cyber sexual harassment. This finding was consistent with previous studies, students in the intervention group showed significant improvement in sexual knowledge as compared with the control group after they underwent a sexual abuse and harassment prevention program [31,32]. Moreover, Garcia and colleagues found that students who received a sex education program had higher knowledge than the control group [33].

At the baseline survey, students in the intervention group tended to have a more negative attitude. The students’ attitudes improved after the educational session. The use of simulated news articles and Facebook messages was successful in addressing and raising participant attitudes about cyber sexual harassment because they were so similar to the participants’ usual online experiences. By emphasizing potential negative effects for the victims, group discussion, empathy training, skill training about online protection strategies, and holding awareness-raising sessions, the program may have a positive influence on attitudes. This was consistent with a previous study on the Media Heroes program for cyberbullying prevention. The program is theoretically based on the theory of reasoned action and the theory of planned behavior. Education implemented by conveying negative consequences for and improving empathy with the victims influences their attitudes in a positive way [34]. Moreover, a study on gender and sexual harassment programs found that students in the treatment group had improved attitudes toward gender and sexual harassment after the program implementation. Hence, school-based intervention programs may improve students’ attitudes through interaction-skill building [35].

The average score of intention to cope with cyber sexual harassment after the intervention and follow-up increased from baseline with a significant difference. The intention to cope with cyber sexual harassment may be increased by implementing a program based on the theory of planned behavior that includes an introductory lecture, peer-led discussions, brainstorming sessions, group activities, and sharing experiences sessions. This was consistent with Lijster et al.’s findings wherein students in the intervention group reported improving behavioral intention to prevent sexual harassment at post-test. [36]. A previous study on sexual abuse prevention with an online program found that at postintervention, students in the treatment group had improved intentions for preventing sexual abuse [37]. Furthermore, a study on the empowering digital citizenship program found that the intervention could influence the students’ anti-cyberbullying intentions [38].

Through program activities, including emotional management, empowering techniques, sharing experiences, slogan creation, and help-seeking processes, students have multiple opportunities to practice their coping skills, which helps to increase their confidence and competence. Several coping strategies were actively used by the students in the intervention group. Students can develop their coping mechanisms by expressing empathy and care for themselves, expressing compassion to peers who have gone through similar things, and fostering help-seeking activities. Students also learned how to recognize symptoms of harmful thinking, such as self-blame, helplessness, and isolation, applying cognitive behavioral therapy. This program may help cyber sexual harassment victims in eliminating self-blame and boosting their confidence to deal with the problem. This was consistent with Kerry et al.’s study wherein their program engaged adolescent victims’ coping skills. Their results showed that students had significant increases in coping skills after completing the psychoeducational program [39]. Furthermore, Machackova et al. found that cyberbullying victims were active in using various methods to cope with the problem [40].

In the present study, the average score of cyber sexual harassment after the intervention and at follow-up decreased from baseline, with a statistically significant difference between the intervention and control groups and measurement times. These findings support the effects of a blended learning program in reducing cyber sexual harassment experiences among female students. The results of this study can be explained in various ways. It is reasonable to assume that specific program components, such as cooperative activity, victim emotional support, peer support, and coping strategies addressing cybervictimization, increased its effects. In this setting, students listen to each other, identify possible harassment circumstances, and support each other during difficult situations. The findings of this study are congruent with that of previous studies on programs to prevent cyberbullying. The use of intervention strategies, such as educating students safe online behavior or how to avoid cyberbullying, have significantly decreased student cybervictimization [41,42,43].

No dropout or loss of student participation in the program occurred in this study. However, not all activities stimulate participation and performance. Students were found to have diverse perspectives when it came to online activities. The effects of program participation may vary according to the content of the interaction, Internet connection, willingness to participate in the discussion, and environmental factors. Blended learning can be made more effective by students’ interactions with their peers and the research team. In blended learning, infographics, videos, graphs, and other types of learning materials are employed. This makes it easier for students to concentrate and integrate information. It enables students to access interactive contents no matter where they are or what they are learning about [29]. Therefore, technological quality should be considered to ensure learning performance in blended learning [44]. Self-report was conducted in the questionnaire, which exposes participants to biases such as recall, social desirability, and response fatigue. A cross-check of self-report information with peer assessment and research team observation would have greatly increased the outcome’s validity. Moreover, it is important to pay attention when students show a significant change in their online and social behavior. Schools also have a responsibility to educate students on how to behave appropriately and safely when using digital media. This involves teaching students on how to identify, react to, and keep away from cyber sexual harassment. Creating a pleasant and safe learning environment in the classroom, integrating teachings on cyber sexual harassment into the curriculum, and advocating for a school digital citizenship program are critical measures to protect students from cyber sexual harassment.

## 5. Conclusions

The results of the present study showed that after female high-school students received a blended learning program, they have improved knowledge, attitude, intention to cope with cyber sexual harassment, and behavioral coping strategies. Furthermore, the blended learning program also decreased cyber sexual harassment experiences among students. To learn more about cyber sexual harassment and victims’ feelings, a mixed methods study and in-depth interviews should be conducted. In addition, applying for this program in other areas or groups, while including a 6- or 12-month follow-up phase to develop a better understanding of intervention effects, sustainability, or long-term maintenance of the program’s beneficial results, is recommended.

## Figures and Tables

**Table 1 ijerph-19-08209-t001:** Overview of program contents.

Week	Concept and Outcomes	Method	Description
1. Getting to know each other	- Icebreaking - Warmup	- Pretest - Create a Facebook group	- The researcher explained the program, and students introduced themselves.
2. Knowing about cyber sexual harassment	- Knowledge improvement	- Group discussion - Lecture in class - Facebook group	- Basic information about cyber sexual harassment was discussed.
3. Realizing risk factors and negative consequences	- Knowledge improvement - Raising an awareness	- Lecture in class - Group discussion - Facebook group	- Risk factors and negative consequences were discussed.
4. What if you are the victim	- Knowledge improvement - Attitudes	- Quizzes - Discussion - Facebook group	- Self-protection strategies and ways to prevent online harassment were discussed.
5. Awareness rising	- Attitude - Emotional management	- Emotional guessing game - Facebook group	- Students worked on enhancing their emotional management skills.
6. Building positive attitudes and girl power	- Knowledge - Theory of empowerment	- Positive message - Discussion - Facebook group	- Empowering students to cope with the problem was implemented.
7. Coping strategies	- Knowledge - Development of coping skills	- Quizzes - Group discussion - Facebook group	- Coping strategies for students were conducted.
8. Taking action and moving forward together	- Knowledge - Theory of empowerment	- Discussion - Lecture - Facebook group	- Students learned the methods to prevent the occurrence of the problem.
9. Lose your fear and find your voice	- Intention to cope with the problem	- Discussion - Facebook group	- Motivated students to report the incident to authorities or consult with someone.
10. Stories worth sharing	- Coping strategies	- Discussion - Sharing experiences - Facebook group	- Students shared their experiences about cyber sexual harassment.
11. Develop campaign slogans	- Intention - Behavioral changes	- Discussion - Create slogans - Facebook group	- Created a message or slogan to encourage others to think before they act online.
12. Summary and evaluation	- Behavioral changes	- Summarize - Posttest	- All activities were summarized.

**Table 2 ijerph-19-08209-t002:** Comparison of sociodemographic characteristics between the intervention and control groups at baseline.

Sociodemographic Characteristics	Intervention Group (*n* = 56)	Control Group (*n* = 56)	*p*-Value
	*n* (%)	*n* (%)	
**GPA**			0.675 ^a^
≤2.00	5 (8.9)	8 (14.3)	
2.01–3.00	29 (51.8)	27 (48.2)	
3.01–4.00	22 (39.3)	21 (37.5)	
**Sexual Orientation**			0.557 ^b^
Heterosexual	48 (85.7)	51 (91.1)	
Homosexual	7 (12.5)	5 (8.9)	
Bisexual	1 (1.8)	0 (0.0)	
**Relationship Status**			0.317 ^a^
Single	25 (44.6)	33 (58.9)	
Committed dating relationship	9 (16.1)	7 (12.5)	
Uncertain dating relationship	22 (39.3)	16 (28.6)	
**Living Status**			0.358 ^a^
Stay with parent	29 (51.8)	34 (60.7)	
Stay with father or mother	23 (41.1)	16 (28.6)	
Stay with relative or guardian	4 (7.1)	6 (10.7)	
**Parents’ Marital Status**			0.489 ^a^
Married/living together	23 (41.1)	27 (48.2)	
Separated (not divorced)	19 (33.9)	20 (35.7)	
Divorced/widowed	14 (25.0)	9 (16.1)	

Analyzed with ^a^ Chi-square test, ^b^ Fisher’s exact test.

**Table 3 ijerph-19-08209-t003:** Comparison of Internet usage between the intervention and control groups at baseline.

Internet Usage	Intervention Group (*n* = 56)	Control Group (*n* = 56)	*p*-Value
	*n* (%)	*n* (%)	
**Duration of Internet Use**			0.151
About 4–5 h per day	7 (12.5)	9 (16.1)	
About 6–7 h per day	19 (33.9)	10 (17.8)	
>7 h per day	30 (53.6)	37 (66.1)	
**Types of Social Media**			0.536
Facebook	40 (71.5)	35 (62.5)	
Line	5 (8.9)	6 (10.7)	
Twitter	6 (10.7)	5 (8.9)	
YouTube	5 (8.9)	10 (17.9)	
**Period Time of Internet Usage**			0.171
15:01–18:00	6 (10.7)	4 (7.1)	
18.01–21:00	42 (75.0)	36 (64.3)	
After 21:00	8 (14.3)	16 (28.6)	
**Parent Supervision**			0.609
Never	31 (55.4)	27 (48.2)	
Hardly	20 (35.7)	21 (37.5)	
Sometimes	5 (8.9)	8 (14.3)	

Analyzed with Chi-square test.

**Table 4 ijerph-19-08209-t004:** Effects of intervention program on knowledge, attitudes, intention, behavioral coping strategies, and cyber sexual harassment.

Variables	Baseline	Week 12	Week 20	*p*-Value		
	Mean (SD)	Mean (SD)	Mean (SD)	Group	Time	Group × Time
**Knowledge**				0.028	0.014	0.022
Intervention	6.61 (1.86)	7.36 (1.53)	7.41 (1.56)			
Control	6.59 (1.87)	6.63 (1.88)	6.61 (1.97)			
**Attitudes**				0.014	0.017	0.003
Intervention	38.36 (5.33)	40.43 (4.73)	40.46 (4.65)			
Control	39.43 (5.26)	38.21 (4.72)	38.25 (4.68)			
**Intention**				0.016	0.003	0.042
Intervention	39.43 (5.26)	41.34 (4.62)	41.84 (4.93)			
Control	38.09 (5.21)	39.45 (5.19)	39.36 (5.26)			
**Behavioral coping strategies**				0.031	0.004	<0.001
Intervention	23.30 (2.18)	24.25 (2.02)	24.14 (2.08)			
Control	23.43 (2.22)	23.34 (2.15)	23.29 (2.24)			
**Cyber sexual harassment**				0.025	<0.001	<0.001
Intervention	39.13 (2.55)	37.02 (3.27)	37.23 (3.14)			
Control	38.91 (2.64)	38.93 (2.56)	38.88 (2.65)			

Analyzed with repeated measures ANOVA. Statistical significance at *p* < 0.05.

## Data Availability

Not applicable.

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
