# Peer review of "Effects of Blended Learning Program for Cyber Sexual Harassment Prevention among Female High School Students in Bangkok, Thailand"

_ijerph, 2022, doi:10.3390/ijerph19138209_

Round 1
Reviewer 1 Report
This is overall an interesting paper, and the authors have made efforts to deliver this innovative research. I would like to share my comments for the authors to consider to further improve this paper.
First, more recent studied need to be reviewed especially for the introduction part. The authors are suggested to talk about more on the momentum of doing this research: what are the negative effects of cyber harassment and why focusing on female students? As lots of harassment are taken out against LGBT males as well. Also, why conducting a blended intervention? What is the purpose of blending online and offline intervention? Any related research in this area?
Second, the authors are suggested to talk more about their intervention program. How do you better align online and offline intervention together.
Third, for the methodology part, why did the authors choose these two schools? What is your rationale for selecting these two schools instead of others please?
For the discussion part, the authors need to further streamline their findings and discussions. The current part is too loose and incoherent, besides, the authors need to interact with the existing literature more in the discussion part.
Reviewer 2 Report
Thank you for your good work.
Based on a quasi-experimental study, this paper evaluated the effects of a blended learning program for cyber sexual harassment prevention among female high school students in Thailand. The program contains the meaning of sexual abuse, physical and psychological changes at puberty, and prevention or coping methods delivered through the presentation, video, and group activities.
The authors found a statistically significant difference between the intervention and control groups and between the times of measurements on knowledge.
Overall, this paper is well-designed empirical research that is presented clearly and concisely.
Having said that, I suggest that the authors mention the current study's shortcomings and limitations in the discussion section.
Additionally, it would be helpful to the readers if the authors offered some recommendations for educational professionals on how to successfully implement classroom programs to minimize cyber-sexual harassment among youths.
Round 2
Reviewer 1 Report
I would suggest the editor to accept this manuscript.